RNA sequencing-based exploration of the effects of blue laser irradiation on mRNAs involved in functional metabolites of D. officinales

Li Hansheng 1
Qiu Yuqiang 2
Sun Gang 3 sungang@nenu.edu.cn
Ye Wei 4 yewei922@qq.com
1 College of Architectural Engineering, Sanming University , Sanming , Chian
2 Xiamen Institute of Technology , Xiamen , China
3 College of Resources and Chemical Engineering, Sanming University , Sanming , China
4 The Institute of Medicinal Plant, Sanming Academy of Agricultural Science , Shaxian , China
Fukushima Atsushi
Electronic publication date: 2022 Jan 4
Publication date: 2022
Volume: 10
Electronic Location ID: e12684
Received 2021 Aug 27; Accepted 2021 Dec 3
Copyright: © 2022 Li et al.
Copyright year: 2022
Copyright holder: Li et al.
License: This is an open access article distributed under the terms of the Creative Commons Attribution License, which permits unrestricted use, distribution, reproduction and adaptation in any medium and for any purpose provided that it is properly attributed. For attribution, the original author(s), title, publication source (PeerJ) and either DOI or URL of the article must be cited.
License URL: https://creativecommons.org/licenses/by/4.0/

Keywords: Dendrobium officinale Kimura et Migo., Blue laser, Functional metabolites, RNA-seq

Funding: National Natural Science Foundation of China 31501802 Natural Science Foundation of Fujian Province 2020J01377 Education research project for young and middle-aged teachers in Fujian JAT190696 Sanming University Scientific Research Foundation for High-level Talent 18YG01, 18YG02, 19YG06 2019 and 2020 Special Commissioner of Science and Technology of Fujian Province This work was funded by the National Natural Science Foundation of China (31501802), the Natural Science Foundation of Fujian Province (2020J01377), the Education research project for young and middle-aged teachers in Fujian (JAT190696), the Sanming University Scientific Research Foundation for High-level Talent (18YG01, 18YG02, 19YG06), and the 2019 and 2020 Special Commissioner of Science and Technology of Fujian Province. The funders had no role in study design, data collection and analysis, decision to publish, or preparation of the manuscript.

==============================
Dendrobium officinale Kimura et Migo (D. officinale) has promising lung moisturizing, detoxifying, and immune boosting properties. Light is an important factor influencing functional metabolite synthesis in D. officinale. The mechanisms by which lasers affect plants are different from those of ordinary light sources; lasers can effectively address the shortcomings of ordinary light sources and have significant interactions with plants. Different light treatments (white, blue, blue laser) were applied, and the number of red leaves under blue laser was greater than that under blue and white light. RNA-seq technology was used to analyze differences in D. officinale under different light treatments. The results showed 465, 2,107 and 1,453 differentially expressed genes (DEGs) in LB-B, LB-W and W-B, respectively. GO, KEGG and other analyses of DEGs indicated that D. officinale has multiple blue laser response modes. Among them, the plasma membrane, cutin, suberine and wax biosynthesis, flavone and flavonol biosynthesis, heat shock proteins, etc. play central roles. Physiological and biochemical results verified that blue laser irradiation significantly increases POD, SOD, and PAL activities in D. officinale. The functional metabolite results showed that blue laser had the greatest promoting effect on total flavonoids, polysaccharides, and alkaloids. qPCR verification combined with other results suggested that CRY DASH, SPA1, HY5, and PIF4 in the blue laser signal transduction pathway affect functional metabolite accumulation in D. officinale through positively regulated expression patterns, while CO16 and MYC2 exhibit negatively regulated expression patterns. These findings provide new ideas for the efficient production of metabolites in D. officinale.

Introduction

Dendrobium officinale Kimura et Migo is a medicinal plant of the Orchidaceae, Dendrobium genus. It can be called a “pulmonary scavenger” because it nourishes and regulates the lungs, produces fluids, moisturizes the lungs, detoxifies, and enhances immunity (Chen et al., 2018). In the fight against the novel coronavirus and the resulting pneumonia, D. officinale is highly favored by Chinese medicine experts. D. officinale is mainly distributed in western Fujian, eastern Zhejiang, southwest Anhui, Sichuan, northwest Guangxi and southeast Yunnan (Xu et al., 2015). D. officinale is susceptible to environmental influences such as light, temperature and humidity, and its natural survival rate is extremely low (Tang et al., 2019). After a long period of uncontrolled mining, wild resources of this species are becoming scarce (Tang et al., 2019). Modern chemical and pharmacological studies have shown that the main chemical components of D. officinale are polysaccharides, flavonoids, bibenzyls and alkaloids (Lin et al., 2019).

Researchers have used different methods to increase the content of functional metabolites in D. officinale, with light regulation being one of the most effective methods. Ordinary light sources refer to all light sources except lasers, including natural light and various artificial light sources (including LEDs). These all produce incoherent light. Additionally, there have been many reports on the study of plant functional metabolites. In a study examining the effect of different LED illumination modes on the accumulation of polysaccharides in D. officinale protocorms, it was found that mixed red and blue light had the best effects. Among the treatments, the yield of polysaccharides was the highest at the red light:blue light ratio of 1:3 (Lin & Lai, 2015). In research on D. officinale with different light qualities, it was found that monochromatic blue light within 15 days was beneficial for increasing the alkaloid content, and monochromatic yellow light within 30 days was beneficial for increasing the alkaloid content. After 30 days, treatment with a ratio of red to blue light of 2:3 was the most conducive to an increase in alkaloid contents (Liu et al., 2016).

To date, there have been few reports on the effects of laser light sources on plant functional metabolites. (1) The interaction between coherent light and plants and the difference between it and incoherent light is a research field with important theoretical value and high levels of innovation. The light emitted by ordinary light sources is composed of wave trains with a length of only a few microns (Tang et al., 2019; Luijtelaar et al., 2019). There is no correlation between the light wave trains, and the interaction between them and plant molecules is also irregular (Tang et al., 2019; Luijtelaar et al., 2019). Laser light sources are composed of several million to hundreds of millions of wave trains of light waves with a length of several meters to several kilometers (Tang et al., 2019; Luijtelaar et al., 2019). There is a strong correlation between the light wave trains. When they interact with plant molecules, they also show a certain regularity (Tang et al., 2019; Luijtelaar et al., 2019). (2) The mechanism of laser is completely different from ordinary light source, it not only can effectively solve the problems of ordinary light sources, but also has significant interaction with plants (Wan, Shi & Zhang, 2020). (3) The laser light source plays an important role in the growth and development of plants. A large number of studies have found that pretreatment of seeds with laser light sources can significantly improve the growth, development and metabolism of a variety of plants (Li, Gao & Han, 2016). For example, when using a 4.0 × 10−3 J/cm2/s laser to treat white lupin and broad bean, after 120 h, it was found that the amylolytic enzyme activity in the seed cells reached the maximum value, the IAA content of the seeds increased, hypocotyl elongation accelerated, and the fresh weight and root length increased (Li, Gao & Han, 2016). Research by Gao, Li & Han (2016) found that a He-Ne laser can accelerate plant growth and development, possibly because it activates the synthesis and release of endogenous nitric oxide (NO) signal molecules and calcium signals in plant cells, which in turn mediates the occurrence of a series of intracellular signal cascade reaction pathways. In summary, the effect of laser light sources on the functional metabolites of D. officinale is worthy of in-depth study.

In the author’s long-term research work on the interaction between ordinary light sources and plants (Li et al., 2019; Li et al., 2021), it was found that ordinary light sources may recognize and act on specific receptors on the cell wall (or cell membrane). The signal transduction pathway activates certain enzymes or substrate proteins, affects specific biochemical and metabolic processes in the cytoplasm, acts on nuclear genes, changes the expression patterns and transcriptional activities of related genes, translates to synthesize specific protein products, and regulates plant growth and development and secondary metabolism. However, the exact molecular mechanism by which lasers regulate plant functional metabolites is unclear. Such regulation may occur through a specific signal transduction pathway or may be caused by the electromagnetic effect of the laser or the energy conversion of laser irradiation or laser coherence (Li, Gao & Han, 2016). These problems urgently need to be further explored and verified.

In this study, we used high-throughput sequencing technology to identify putative mRNAs and investigated their expression profiles in D. officinale under different light patterns. By comparing and analyzing the sequencing data of the treatment group and the control group, the difference between the functional metabolites of D. officinale under normal and laser light sources was found, the specific effects of the laser on mRNA and secondary metabolites of D. officinale were discovered, the secondary metabolites of D. officinale were mapped, and their signal transduction pathways were determined. These results will provide new ideas for the high-yield production of medicinal ingredients of D. officinale.

Materials and Methods

Plant materials and light treatments

The D. officinale used in this study had 3 to 4 true leaves, a leaf width of approximately 2 to 3 mm, and a seedling height of approximately 2 cm. The following light and environmental parameters were used: blue laser (450 nm), blue light (450 nm), white light; total light intensity (100/umol•m−2•S−2); light time (12 h/d); humidity 55%~60%; and temperature 26–28 °C. Tissue cultured seedlings of D. officinale were placed in a light incubator for 60 days. Tissue culture seedlings of D. officinale were maintained as previously described (Li et al., 2021). The control and light treatment samples were stored for subsequent nucleic acid extraction, high-throughput sequencing and functional metabolite content determination. White light was used as the control group, and the experimental group included blue light and blue laser treatments. Samples treated with blue laser vs blue light, blue laser vs white light and blue light vs white light were named BL-B, BL-W, and B-W, respectively.

mRNA library construction and Illumina HiSeq sequencing

In this study, high-throughput sequencing was performed on samples subjected to three treatments-white light, blue light, and blue laser irradiation-with three biological replicates for each group. RNA quantification and qualification were performed according to the method of Li et al. (2019, 2021). Sequencing library preparation and high-throughput sequencing were subsequently performed using the Illumina HiSeq platform (Beijing, China). All sequencing data of D. officinale under the different light treatments were deposited in the National Genomics Data Center (NGDC) Sequence Read Archive (accession number PRJCA006154).

The adaptor sequences and low-quality sequence reads were removed from the data sets. Raw sequences were transformed into clean reads after data processing. These clean reads were then mapped to the reference genome sequence (Zhang et al., 2016). The reference genome version of D. officinale in this manuscript was updated on April 11, 2019 (Li et al., 2021). Only reads with a perfect match were further analyzed and annotated based on the reference genome.

Gene functional annotation and differential expression analysis

Gene function was annotated as previously described (Li et al., 2021). Differential expression analysis of two conditions/groups was performed using the DESeq R package (1.10.1). DESeq provides statistical tools for determining differential gene expression using a model based on the negative binomial distribution. The resulting P-values were adjusted using Benjamini and Hochberg’s approach for controlling the false discovery rate (Benjamini & Yekutieli, 2005). Genes with an adjusted P-value < 0.01 and absolute value of log2(Fold change) >1 found by DESeq were assigned as differentially expressed.

GO and KEGG enrichment analyses of differentially expressed genes

Gene Ontology (GO) enrichment analysis of the differentially expressed genes (DEGs) was implemented by the GOseq R package-based Wallenius noncentral hypergeometric distribution (Young et al., 2020), which can adjust for gene length bias in DEGs. KEGG (Kyoto Encyclopedia of Genes and Genomes) enrichment analysis of the DEGs was performed using KOBAS (Chen et al., 2011) software.

Determination of functional metabolites

The polysaccharide, flavonoid and alkaloid contents in the D. officinale stems and leaves were determined as previously described (Li et al., 2021).

The anthocyanin of D. officinale stems and leaves were determined by the method described by Chen & Department (2016). Two grams of Freeze-dried grains was weighed and added to 20 mL of extraction solution (1% vanillin methanol solution: 15% methyl hydrochloride alcohol solution = 1:1). Then dilute the volume to 50 mL with the extraction solution and let it stand for 2 h. Take 1 mL of supernatant in a 250 mL erlenmeyer flask, add 100 mL of 1% hydrochloric acid solution, and water bath for 10 min (60 °C). The absorbance was measured with a UV-visible spectrophotometer, and the wavelengths of anthocyanins were measured at 520 nm. The anthocyanin contents in the D. officinale stems and leaves were calculated according to established standard curves.

Determination of enzyme activity

The leaves of D. officinale were ground with liquid nitrogen, 0.2 g grains were weighed, and 1 ml extract was added. Then, the extract was centrifuged (4 °C, 8,000 g, 10 min), and the supernatant was collected and placed on ice for testing. The supernatant was collected for enzyme activity assays. Superoxide dismutase (SOD), peroxidase (POD) and Phenylalaninammonialyase (PAL) were assayed using commercial kits (Suzhou Keming Biotechnology Limited Company, Suzhou, China) and a DU640 spectrophotometer (Beckman, Brea, CA, USA) according to the manufacturers’ instructions and a previous report.

Quantitative real-time PCR (qRT-PCR) analysis

Total RNA from D. officinale leaves was used for qRT-PCR validation of mRNAs. Twelve DEGs were subjected to qRT-PCR analysis on a LightCycler 480 Real-Time PCR System (Roche, Basel, Switzerland). cDNA synthesis, the reaction system and the procedures, etc. referred a previous method (Li et al., 2021). The relative mRNA expression levels were calculated using the comparative 2−ΔΔCt method. The ACTIN gene was used as the reference gene (Shen et al., 2017). The primer sequences are listed in Table S1.

Data analysis

The quantitative results of gene expression, antioxidant enzyme activity and functional metabolites of D. officinale were determined with at least three biological replicates. Using SPSS V 19.0 and one-way analysis of variance (ANOVA) with Duncan’s tests, the impact of different illumination modes on the indicators of D. officinale was analyzed. GraphPad Prism 6.0 software and OmicShare online software were used for drawing.

Results

Growth state of D. officinale under different light treatments

Lasers can affect the growth state of D. officinale. The results of this study found that there were no significant differences in plant height, stalk thickness, and leaf area between white light, blue light and blue laser, but the number of red leaves under the blue laser treatment was greater than that under the blue and white light treatments (Figs. 1A–1F; Table S2).

Figure 1 The phenotype of D. officinale under different light treatments.

(A–C) The phenotype before white, blue and blue laser light treatments respectively. (D–F) The phenotype after white, blue and blue laser light treatments respectively. Bars = 10 mm.

Sequencing and assembly of transcriptomic data

To study the effects of lasers on functional metabolite-related genes of D. officinale, 9 mRNA libraries were constructed and sequenced in this study (Table 1). After removing the linker sequences, the RNA-seq data of D. officinale under different light treatments produced 40,640,606 to 54,223,452 reads. A total of 87.96–88.97% of clean reads could be completely matched to the reference genome of D. officinale, while 85.70–86.70% of clean reads could be matched to a single site in the reference genome of D. officinale, indicating that sequencing reads matched the reference genome of D. officinale to a high degree. The Q30 values of D. officinale samples were all higher than 92.47%, indicating the high reliability of the D. officinale transcriptome sequencing data.

Table 1 mRNA results from nine D. officinale libraries.

Samples	Total reads	Total mapped reads (%)	Uniquely mapped reads (%)	Q30 (%)	GC content (%)	
W1	50,264,012	88.63	86.20	92.53	46.07	
W2	53,636,818	88.86	86.56	92.75	46.13	
W3	54,223,452	88.97	86.42	92.47	46.03	
B1	45,129,776	89.02	86.70	93.93	46.02	
B2	40,640,606	88.54	86.17	93.72	45.95	
B3	42,995,976	87.96	85.70	93.41	45.91	
LB1	41,961,634	88.76	86.44	93.12	45.83	
LB2	50,932,878	88.42	86.10	92.54	45.78	
LB3	42,002,508	88.40	86.16	92.63	45.86	

Analysis of differentially expressed genes under different light treatments

This study analyzed the results of transcriptome sequencing and discovered 3,735 new genes, 2,888 that were functionally annotated, and 2,500 DEGs (Table S3). To study the gene expression of D. officinale, this study divided all genes into three categories, namely, high expression (FRKM > 50), medium expression (5 ≤ FRKM ≤ 50) and low expression (FRKM < 5). The results showed that most genes were in the low and medium expression groups, while a few genes were highly expressed (Fig. 2A).

Figure 2 Differentially expressed genes under the different light treatments.

(A) Number of genes with low (FPKM < 5), moderate (5 ≤ FPKM ≤ 50) and high (FPKM > 50) expression in each library. (B) Venn diagram showing the numbers of unique and commonly regulated genes identified from the LB-B, LB-W and B-W comparisons. (C) DEGs in response to different light treatments.

To study the DEGs of D. officinale, we drew Venn diagrams of DEGs under different combinations. The results showed that 465 DEGs were identified in the BL-B combination, 2,107 were identified in the BL-W combination, and 1,453 were identified in the B-W combination. Eighty genes were jointly regulated by the three combinations of BL-B, BL-W and B-W; 141 genes were jointly regulated by the two combinations of BL-B and BL-W; 325 genes were jointly regulated by the two combinations of BL-B and B-W; and 1,139 genes were jointly regulated by the two combinations of BL-W and B-W. Seventy-nine genes were specifically regulated by the BL-B combination, 253 genes were specifically regulated by the BL-W combination, and 723 genes were specifically regulated by the B-W combination (Fig. 2B). Cluster analysis showed that 2,500 DEGs could be divided into six expression patterns under different light treatments (Fig. 2C).

GO analysis of DEGs in D. officinale

To further understand the influence of laser irradiation on D. officinale, GO enrichment analysis was performed on the DEGs in the three comparisons of BL-B, BL-W, and B-W (Table 2).

Table 2 Top five GO terms obtained from an enrichment analysis of DEGs in D. officinale under different light treatments.

	ID	Description	q value	
LB-B	LB-W	B-W	
Biological
process	GO:0006614	SRP-dependent cotranslational protein targeting to membrane	0.003			
GO:0043067	regulation of programmed cell death	0.0035	0.0027	0.00017	
GO:0043254	regulation of protein complex assembly	0.004			
GO:0006446	regulation of translational initiation	0.0056			
GO:0034976	response to endoplasmic reticulum stress	0.0072			
GO:0051130	positive regulation of cellular component organization		0.002	0.00161	
GO:0015807	L-amino acid transport			0.00353	
GO:0009813	flavonoid biosynthetic process			0.00407	
GO:0008154	actin polymerization or depolymerization		0.008	0.00627	
GO:0009694	jasmonic acid metabolic process		0.0017		
GO:0031408	oxylipin biosynthetic process		0.0027		
Cellular
component	GO:0005786	signal recognition particle, endoplasmic reticulum targeting	0.0016	0.0038	0.0028	
GO:0044459	plasma membrane part	0.0113			
GO:0080008	Cul4-RING E3 ubiquitin ligase complex	0.0115	0.0025	0.0165	
GO:0005773	vacuole	0.0128			
GO:0030894	replisome	0.0149			
GO:0031226	intrinsic component of plasma membrane			0.0017	
GO:0016021	integral component of membrane		0.0033	0.0043	
GO:0005887	integral component of plasma membrane		0.0091	0.0174	
GO:0005618	cell wall		0.0148		
Molecular
function	GO:0008312	7S RNA binding	0.00044	0.00183	0.00119	
GO:0016702	oxidoreductase activity, acting on single donors with incorporation of molecular oxygen, incorporation of two atoms of oxygen	0.00064	0.00111	0.00174	
GO:0051082	unfolded protein binding	0.00138			
GO:0016705	oxidoreductase activity, acting on paired donors, with incorporation or reduction of molecular oxygen	0.00438	0.00012	0.00062	
GO:0003964	RNA-directed DNA polymerase activity	0.00536			
GO:0016210	naringenin-chalcone synthase activity		0.00056	0.0003	
GO:0005506	iron ion binding		0.00032	0.00033	

For the biological process category in the GO analysis, the BL-B-specific terms included SRP-dependent cotranslational protein targeting to membrane, regulation of translational initiation, regulation of protein complex assembly, response to endoplasmic reticulum stress, etc. The BL-W-specific biological processes included jasmonic acid metabolic process and oxylipin biosynthetic process.

For the cell component category in the GO analysis, the BL-B-specific terms included plasma membrane part, vacuole, and replisome, and only cell wall was specific to BL-W.

For the molecular function category in the GO analysis, the BL-B-specific terms included unfolded protein binding and RNA-directed DNA polymerase activity.

In summary, the effect of laser irradiation on the functional metabolites of D. officinale might be closely related to SRP-dependent cotranslational protein targeting to the membrane, jasmonic acid metabolic process, oxylipin biosynthetic process, plasma membrane, cell wall, etc.

KEGG enrichment analysis of DEGs in D. officinale

This study included KEGG enrichment analysis of the identified DEGs (Fig. 3). The top five enriched pathways were cutin, suberine and wax biosynthesis, vitamin B6 metabolism, photosynthesis-antenna proteins, flavone and flavonol biosynthesis, and diterpenoid biosynthesis for the BL-B combination (Fig. 3A). The top five enriched pathways were plant hormone signal transduction, flavonol and flavonol biosynthesis, vitamin B6 metabolism, cutin, suberine and wax biosynthesis, and circuitadian rhythm-plant for the BL-W combination (Fig. 3B). The top five enriched pathways were flavone and flavonol biosynthesis, flavonoid biosynthesis, stilbenoid, diarylheptanoid and gingerol biosynthesis, monobactam biosynthesis, and alpha-linolenic acid metabolism for the B-W combination (Fig. 3C).

Figure 3 KEGG enrichment analysis of DEGs in D. officinale under different light treatments.

(A) LB-B; (B) LB-W. (C) B-W; (D) Top 20 KEGG pathways enriched in DEGs in the three groups. The red colour indicates that the comparison contains the pathway, and the blue colour indicates that the comparison does not contain the pathway.

This study identified the top 20 enrichment pathways for the 3 combinations (LB-B, LB-W, BW), including flavonol and flavonol biosynthesis, flavonoid biosynthesis, circadian rhythm – plant, stilbenoid, diarylheptanoid and gingerol biosynthesis, phenylpropanoid biosynthesis, glutathione metabolism, cutin, suberine and wax biosynthesis, indicating that these pathways were significantly different under the three illumination modes of white light, blue light, and blue laser (Fig. 3D).

Plant-pathogen interaction, cysteine and methionine metabolism, sphingolipid metabolism, plant hormone signal transduction, and alpha-linolenic acid metabolism were among the top 20 most enriched pathways in the LB-W and BW combinations (Fig. 3D). Taurine and hypotaurine metabolism, fructose and mannose metabolism, vitamin B6 metabolism, and diterpenoid biosynthesis were among the top 20 most enriched pathways for the combination of LB-B and LB-W (Fig. 3D). The above results indicate that the blue laser plays an important role in these pathways.

Some pathways were only among the top 20 most enriched pathways for one combination; these included pentose and glucuronate interconversions, sulfur metabolism, starch and sucrose metabolism, and protein processing in the endoplasmic reticulum for the LB-B combination (Fig. 3D), indicating that the blue laser has a higher effect on these pathways than blue light. Tropane, piperidine and pyridine alkaloid biosynthesis, galactose metabolism, and phenylalanine metabolism were only among the top 20 most enriched pathways for the LB-W combination (Fig. 3D), indicating that the blue laser has a higher effect on these pathways than white light.

The top 10 up- and downregulated DEGs under different light treatments

In this study, the top 10 up- and downregulated DEGs under the different light treatments were analyzed. The top 10 upregulated genes mainly included heat shock proteins (HSP70, HSP23, HSP18.6, HSP83A, CCOMT, ABCG11) and polysaccharide metabolism and synthesis genes (INV*DC4, CSLA9) in the LB-B combination (Table 3). The top 10 downregulated genes mainly included chlorophyll synthesis-related genes (CAB3C), isoflavone synthesis-related genes (CYP81E1, CYP81E1), and serine-related proteases (HT1, SAT2) in the LB-B combination (Table 3). The top 10 upregulated genes mainly included early light-inducible protein (ELIP1), transcription factor (MYB114), polysaccharide metabolism synthesis gene (MAN6), and heat shock protein (HSP83A, HSP18.6) in the LB-W combination (Table 4). The top 10 downregulated genes mainly included cytochrome-related genes (CYP94A1), ethylene-responsive transcription factors (ERF112, ERF114), and isoflavone synthesis-related genes (CYP81E1) in the LB-W combination (Table 4).

Table 3 The top 10 up-and down-regulation DEGs in the LB-B.

NO	Gene ID	gene_name	log2 (Laser/Blue)	regulated	NR_annotation	
1	gene-MA16_Dca014371	CLPB1	2.946278	up	chaperone protein ClpB1	
2	gene-MA16_Dca006163	HSP70	2.937094	up	heat shock cognate 70 kDa protein 2-like	
3	gene-MA16_Dca000936	INV*DC4	2.902941	up	Beta-fructofuranosidase, soluble isoenzyme I	
4	gene-MA16_Dca018514	CSLA9	2.900806	up	Glucomannan 4-beta-mannosyltransferase 9	
5	gene-MA16_Dca013846	HSP23	2.662168	up	small heat shock protein, chloroplastic-like	
6	gene-MA16_Dca026995	HSP18.6	2.516762	up	18.6 kDa class III heat shock protein	
7	gene-MA16_Dca018083	HSP83A	2.485551	up	heat shock protein 83	
8	gene-MA16_Dca001614	CCOMT	2.662168	up	small heat shock protein, chloroplastic-like	
9	gene-MA16_Dca005188	LIS	2.516762	up	18.6 kDa class III heat shock protein	
10	gene-MA16_Dca001256	ABCG11	2.485551	up	heat shock protein 83	
1	gene-MA16_Dca003810	TSPO	−2.9122	down	translocator protein homolog	
2	gene-MA16_Dca022104	CAB3C	−2.90546	down	chlorophyll a-b binding protein of LHCII type 1-like	
3	gene-MA16_Dca024558	PETE	−2.35939	down	plastocyanin-like	
4	gene-MA16_Dca017227	XERO1	−2.27399	down	Dehydrin Xero 1	
5	gene-MA16_Dca008991	CYP81E1	−2.12773	down	Isoflavone 2&apos;-hydroxylase	
6	gene-MA16_Dca009695	HT1	−2.00848	down	serine/threonine-protein kinase STY46-like isoform X2	
7	gene-MA16_Dca015158	CYP71A1	−2.00817	down	Cytochrome P450 71A1	
8	gene-MA16_Dca016021	GMPM1	−1.99738	down	11 kDa late embryogenesis abundant protein-like	
9	gene-MA16_Dca002504	SAT2	−1.93201	down	probable serine acetyltransferase 2 isoform X1	
10	gene-MA16_Dca008990	CYP81E1	−1.927	down	Isoflavone 2&apos;-hydroxylase	

Table 4 The top 10 up-and down-regulation DEGs in the LB-W.

NO	Gene ID	gene_name	log2 (Laser/Blue)	regulated	NR_annotation	
1	gene-MA16_Dca024845	ELIP1	5.335425	up	early light-induced protein 1, chloroplastic-like	
2	gene-MA16_Dca003829	C1	5.219541	up	trichome differentiation protein GL1-like	
3	gene-MA16_Dca022385	RALFL33	4.06727	up	Protein RALF-like 33	
4	gene-MA16_Dca003827	C1	3.990023	up	transcription factor MYB114-like	
5	gene-MA16_Dca001256	ABCG11	3.550505	up	ABC transporter G family member 11-like	
6	gene-MA16_Dca021634	MAN6	3.505578	up	mannan endo-1,4-beta-mannosidase 6-like	
7	gene-MA16_Dca005399	APG	3.429596	up	GDSL esterase/lipase APG	
8	gene-MA16_Dca018083	HSP83A	3.384163	up	heat shock protein 83	
9	gene-MA16_Dca012014	NIP2-1	3.344414	up	aquaporin NIP2-1-like	
10	gene-MA16_Dca026995	HSP18.6	3.318636	up	18.6 kDa class III heat shock protein	
1	gene-MA16_Dca015278	CYP94A1	−7.75405	down	cytochrome P450 94B3-like	
2	gene-MA16_Dca015280	CYP94A1	−7.31088	down	cytochrome P450 94B1-like	
3	gene-MA16_Dca022981	SRG1	−7.01481	down	probable 2-oxoglutarate-dependent dioxygenase At5g05600	
4	gene-MA16_Dca017290	ABCG11	−6.7603	down	ABC transporter G family member 11	
5	gene-MA16_Dca017057	ERF112	−6.4656	down	Ethylene-responsive transcription factor ERF112	
6	gene-MA16_Dca008990	CYP81E1	−6.2956	down	Isoflavone 2&apos;-hydroxylase	
7	gene-MA16_Dca008681	ERF114	−6.06792	down	ethylene-responsive transcription factor ERF110-like	
8	gene-MA16_Dca014747	CBSX1	−5.90332	down	CBS domain-containing protein CBSX1, chloroplastic-like	
9	gene-MA16_Dca026750	CYP735A1	−5.82538	down	Cytokinin hydroxylase	
10	gene-MA16_Dca015945	FLS	−5.82253	down	probable 2-oxoglutarate-dependent dioxygenase At5g05600	

Levels of physiological and biochemical indicators in D. officinale under different light treatments

In this study, the physiological and biochemical indexes of D. officinale leaves were measured under different light treatments, and the results are shown in Fig. 4. The POD enzyme activity value under the blue laser was 1,866.67 U g−1, blue light was 1,713.33 U g−1, and white light was 760.00 U g−1 (Fig. 4A; Table S5). The SOD enzyme activity value under blue laser treatment was 50.15 U g−1, blue light was 18.90 U g−1, and white light was 8.49 U g−1 (Fig. 4B; Table S6). The PAL enzyme activity under the blue laser treatment was 182.89 U g−1, blue light was 173.03 U g−1, and white light was 50.29 U g−1 (Fig. 4C; Table S7). The activities of POD, SOD and PAL in D. officinale were the highest under the blue laser treatment, followed by blue light, and the lowest under white light. Therefore, blue laser is most beneficial for promoting the activities of POD, SOD and PAL in D. officinale.

Figure 4 Levels of physiological and biochemical indicators in the leaves of D. officinale under different light treatments.

(A) POD activity; (B) SOD activity; (C) PAL activity. Different upper/lowercase letters indicate statistically significant differences at the 0.01/0.05 level, as determined by one-way ANOVA and Duncan’s test.

Secondary metabolite contents in D. officinale under different light treatments

In this study, the functional metabolites of D. officinale leaves and stems were measured under different light treatments, and the results are shown in Fig. 5. Among the differentially expressed flavonoid metabolic pathway synthesis genes, the expression levels of most genes under blue laser treatment were higher than those under the other treatments (Fig. 5A). The flavonoid content of D. officinale leaves under blue laser irradiation was the highest at 73.11 mg/g, followed by that under blue light at 52.55 mg/g and white light at 29.90 mg/g (Fig. 5B; Table S8). The flavonoid content of D. officinale stems under blue laser irradiation was the highest at 41.38 mg/g, followed by blue light at 30.83 mg/g and white light at 22.60 mg/g (Fig. 5C; Table S9). Among the differentially expressed anthocyanin metabolic pathway synthesis genes, the expression levels of most genes under blue laser treatment were higher than those under the other treatments (Fig. 5A). The anthocyanin content of D. officinale leaves under blue laser irradiation was the highest at 20.66 mg/g, followed by that under blue light at 16.82 mg/g and white light at 15.07 mg/g (Fig. 5D; Table S10). The flavonoid content of D. officinale stems under blue laser irradiation was the highest at 9.23 mg/g, followed by blue light at 9.17 mg/g and white light at 8.43 mg/g (Fig. 5E; Table S11). Among the differentially expressed synthetic genes in the polysaccharide metabolic pathway, the expression levels of all genes under the blue laser treatment were higher than those under the other treatments (Fig. 5F). The leaves and stems had the highest polysaccharide contents under blue laser treatment, followed by blue light, and white light resulted in the lowest polysaccharide content (Figs. 5G, 5H; Tables S12, S13). Among the differentially expressed synthetic genes in the alkaloid metabolic pathway, the expression of PPO and TTA genes under blue laser treatment was higher than that under other treatments (Fig. 5I). The leaf alkaloid content was the highest under blue laser treatment at 73.04 mg/g, that under blue light treatment was 72.55 mg/g, and that under white light treatment was the lowest (Fig. 5J; Table S14). The highest alkaloid content of stems under blue laser treatment was 34.29 mg/g, under blue light treatment the content was 34.62 mg/g, and under white light treatment the contents were the lowest (Fig. 5K; Table S15). Therefore, compared with ordinary light sources, blue lasers can significantly increase the contents of total flavonoids, polysaccharides, and alkaloids in D. officinale.

Figure 5 DE metabolic pathway synthetic genes and functional metabolite contents in D. officinale under different light treatments.

(A, F, I) FPKM value changes in DE synthetic genes of flavonoid metabolic pathway, polysaccharide metabolic pathway and alkaloid metabolic pathway respectively. (B and C) Changes in the flavonoid contents in the leaves and stems respectively. (D and E) Changes in the anthocyanin contents in the leaves and stems respectively. (G and H) Changes in the polysaccharide contents in the leaves and stems respectively. (J and K) Changes in the alkaloid contents in the leaves and stems respectively.

Identification of DEGs in D. officinale under different light treatments by qRT-PCR

In this study, nine groups of mRNAs were verified by qRT-PCR, and the results are shown in Fig. 6. Some heat shock proteins, cutin, suberine and wax biosynthesis, and hormone signal transduction pathway-related genes play an important role in blue laser-mediated regulation of functional metabolites of D. officinale. For example, the expression level of HSP70 was highest under blue laser treatment, followed by white light, and was the lowest under blue light (Fig. 6A; Table S14). The expression levels of CYP86A4S and ERF13 were the highest under blue laser irradiation, followed by blue light and white light (Figs. 6B, 6C; Table S14). Blue laser signal transduction pathway genes are closely related to the synthesis of functional metabolites of D. officinale. For example, the expression levels of CRY DASH, SPA1, and HY5 were the highest under blue laser irradiation, followed by blue light, and white light (Figs. 6D–6F; Table S14). The expression levels of CO16 and MYC2 were highest under white light, followed by blue light, and blue laser light had the lowest expression (Figs. 6G, 6H; Table S14). The expression level of PIF4 was highest under blue laser treatment, followed by white light, and was the lowest under blue light (Fig. 6I; Table S14). Therefore, CRY DASH, SPA1, HY5, and PIF4 in the blue laser signal transduction pathway might affect the accumulation of functional metabolites of D. officinale through positively regulated expression patterns, while CO16 and MYC2 exhibit negatively regulated expression patterns.

Figure 6 qPCR verification of DEGs in D. officinale under different light treatments.

(A) HSP70, heat shock cognate 70 kDa protein 2-like; (B) CYP86A4S, fatty acid omega-hydroxylase; (C) ERF13, ethylene response factor 13; (D) CRY DASH, cryptochrome dash; (E) SPA1, phytochrome A suppressor 1; (F) HY5, long hypocotyl 5; (G) CO16, constants 16; (H) MYC2, basic helix-loop-helix transcription factor; (I) PIF4, Phytochrome-interacting factor 4.

Discussion

Cell signaling perception and conduction in D. officinale under blue laser

GO analysis identified SRP-dependent cotranslational protein targeting the membrane, regulation of translational initiation, regulation of protein complex assembly, response to endoplasmic reticulum stress, plasma membrane part, and vacuole as significantly enriched in only the LB-B combination (Table 2). Oxylipin biosynthetic process and the cell wall were significantly enriched in only the LB-W combination (Table 2). All of the above pathways involve cell signaling perception and conduction in D. officinale under blue laser treatment.

When D. officinale responds to a blue laser, the cell wall composition changes accordingly. The sensor elements of external factors are mainly distributed between the cell wall and the cell membrane, leading to an increase in the concentration of Ca2+ in the cytosol (Hamann, 2015). The cell membrane is a barrier that prevents external substances from freely entering the cell and is the site of information, energy, and material exchange between external factors and the cell (Morales-Cedillo et al., 2015). The cell membrane maintains the stability of the intracellular environment so that the physiological metabolic pathways in the plant proceed in an orderly manner. The endoplasmic reticulum is the cytoplasmic membrane system and is connected to the cell membrane on the outside and communicates with the outer membrane of the nuclear membrane on the inside. The endoplasmic reticulum organically connects the various structures in the cell into a whole, effectively increases the membrane area in the cell, and plays the role in external signal transmission and intracellular material transport (Wang, Hawes & Hussey, 2017). Ribosomes are attached to the rough endoplasmic reticulum, and their arrangement is relatively smooth. The endoplasmic reticulum is smooth, and its function is to synthesize protein macromolecules and transport them out of the cell or to other parts in the cell. Oxylipin is a metabolite of oxidized fatty acids and their derivatives (Kriechbaumer & Brandizzi, 2020). Such substances are found in bacteria, fungi, algae, and flowering plants. Oxylipin is a signaling molecule that regulates plant growth and development and plays an important role in responding to external factors (Satoh et al., 2014; Savchenko, Zastrijnaja & Klimov, 2014).

Therefore, when the cell wall of D. officinale is affected by blue laser, the composition and structure of the cell wall dynamically changes to maintain the integrity of the cell wall and adapt to cell growth. The protein on the cell wall and the outer side of the membrane is the first sensor element to sense the blue laser. Together with the receptor protein, these sensing elements initiate a blue laser response, transmit information to the endoplasmic reticulum of D. officinale through a signal cascade, and then regulate the expression of cell wall components and extracellular proteins through a feedback mechanism.

D. officinale responds to blue lasers through cutin, suberine and wax biosynthesis

This study found that cutin, suberine and wax biosynthesis was among the top 20 most enriched pathways for the three combinations (LB-B, LB-W, B-W) by KEGG analysis (Fig. 3). The stratum corneum is a lipid water-retaining layer formed on the outer surface of the epidermal cell wall of terrestrial plants. The basic function of the stratum corneum is to retain water, and it also plays a role in external factor response, self-cleaning, and organ development (Ingram & Nawrath, 2017). The stratum corneum is usually composed of cutin and wax. Cutin is the main structural component of the stratum corneum, and its main component is polyester. The waxy components are mainly very long-chain saturated fatty acids and their derivatives, as well as flavonoids and triterpenoids (Duan, Wang & Chen, 2017). These components are synthesized on the endoplasmic reticulum and transported to the cell surface to further form a complete stratum corneum structure. Relevant studies have found that plant epidermal wax reflects ultraviolet radiation and visible light, and wax most strongly reflects ultraviolet radiation than visible light (Bruhn et al., 2014). Some studies also found that compared with normal plants, corn lacking wax in the epidermis suffered more damage from ultraviolet rays, and leaf shape and plant genetics were significantly affected (Duan, Wang & Chen, 2017). Compared with leaves with relatively less epidermal wax, leaves with more epidermal wax absorbed more ultraviolet rays, with reduced damage to the plant (Vishwanath et al., 2015). In addition, suberin is a glycerol-phenol-lipid polymer whose composition is similar to that of cell wall wax. Suberin can control the outflow of water and solutes, and it can also play an important role in the resistance of plants to external factors such as drought, salt and strong light (Vishwanath et al., 2015; Martins et al., 2014). Therefore, long-term cultivation of D. officinale under an appropriate amount of blue laser (strong radiation energy) promotes the formation of plant wax, cutin, and cork, ensures the normal growth and development of plants, and increases the accumulation of functional metabolites.

Heat shock proteins play an important role in the response of D. officinale to blue lasers

This study found that the top 10 differentially expressed genes in the LB-B and LB-W combinations involved heat shock protein-related genes, including HSP70, HSP23, HSP18.6, HSP83A, CCOMT, and ABCG11 (Tables 3, 4). This study also found that the activities of POD and SOD were higher under blue laser than under blue and white light (Figs. 4A, 4B). GO analysis identified the oxidoreductase activity, acting on single donors with incorporation of molecular oxygen, incorporation of two atoms of oxygen and oxidoreductase activity, acting on paired donors, with incorporation or reduction of molecular oxygen as significantly enriched in three combination (Table 2). Heat shock proteins play an important role in the growth of plants, as both essential and defensive proteins, helping to rebuild the normal structure and function of cells (Jacob, Hirt & Bendahmane, 2017). When plants are under adverse stress, heat shock proteins mainly participate in the defencse response in the following ways. (1) Heat shock proteins can protect protein functional structure, prevent abnormal protein aggregation, and remove potentially harmful denatured proteins (Jacob, Hirt & Bendahmane, 2017; Fu & Zou, 2015). (2) Heat shock proteins maintain membrane integrity by increasing membrane lipid order and reducing membrane lipid fluidity (Jacob, Hirt & Bendahmane, 2017; Fu & Zou, 2015). (3) Heat shock proteins act as antioxidants to remove excess reactive oxygen species (ROS). Plants have a complete internal protective enzyme system to remove the damage caused by ROS, thereby maintaining the normal functioning of plant cells. Plant cells mainly use enzymatic and nonenzymatic antioxidant systems to reduce oxidative damage. The compounds in the enzymatic system include superoxide dismutase (SOD), peroxidase (POD), and catalase (CAT), etc (Luo et al., 2020). Compounds in the nonenzymatic system include flavonoids, ascorbate, glutathione, etc (Luo et al., 2020). The increase in plant ROS levels can cause oxidative stress, which in turn causes oxidative damage to nucleic acids, proteins, carbohydrates and lipids. Heat shock proteins may increase glutathione by enhancing the activity of glucose-6-phosphate dehydrogenase (G6PD). The reduction of peptides (glutathione, GSH) eliminates excess ROS (Liu et al., 2016). Therefore, heat shock proteins might play an important role in the response of D. officinale to blue lasers.

Some functional metabolic pathways are involved in the effect of blue laser irradiation on D. officinale

This study found that the PAL enzyme activity under blue laser treatment was significantly higher than that under blue and white light treatments (Fig. 4C). PAL is the key and rate-limiting enzyme in the phenylpropane metabolic pathway. Phenylalanine is converted into cinnamic acid under the catalysis of the PAL enzyme. An increase in its activity can produce lignin, cork, flavonoids and anthocyanins, thereby improving the ability to deal with external factors. In the D. officinale response to blue laser irradiation, this study also found that phenylalanine metabolism, flavone and flavonol biosynthesis, starch and sucrose metabolism, fructose and mannose metabolism, tropane, piperidine and pyridine alkaloid biosynthesis and other pathways were among the top 20 most enriched pathways (Fig. 3), and these pathways are closely related to the biosynthesis of flavonoids, polysaccharides, and alkaloids. The above results indicate that the blue laser promotes the accumulation of total flavonoids, polysaccharides and alkaloids in D. officinale.

Light is an indispensable factor for plant growth and development. Plants form their own unique metabolic basis by sensing different light signals (Nhut et al., 2015). This study found that the PAL enzyme activity under blue laser treatment was significantly higher than that under blue and white light treatments. Flavonoids in plants have the protective function of scavenging ROS, and light will cause different degrees of photooxidative stress, thereby inducing the ROS scavenging mechanism and affecting the accumulation of flavonoids (Lan et al., 2017; Martin et al., 2015). This effect may have been due to the high-energy blue laser used, which caused greater photooxidative damage to D. officinale, and flavonoids played a role in protecting plants by removing ROS. The synthesis of plant alkaloids is mainly affected by genetic factors and the external environment (Wang et al., 2010). The external factors that affect the synthesis of plant alkaloids mainly include strong and weak light stress, temperature and drought (Wang et al., 2010). Light can promote an increase in secondary messengers (calmodulin, G protein and cAMP contents), which in turn activates the phytochrome and cryptochrome system and enables the expression of downstream genes related to alkaloid metabolism in plants (Wang et al., 2010). In this study, the blue laser may promote an increase in the secondary messenger content, thereby increasing their contents in D. officinale. Plants can survive under adverse conditions such as ultraviolet light, strong light, low temperature, and high salt because they secrete long-chain polysaccharides, that is, extracellular polysaccharides (EPSs), outside the cell during growth and metabolism. On the one hand, EPSs play a role as osmotic regulators; on the other hand, they can form a protective film on the cell surface to protect proteins from inactivation (Chang, Chen & Gong, 2020). In summary, the metabolism of flavonoids, alkaloids, and polysaccharides plays an important role in the response of D. officinale to blue lasers.

The blue laser signal transduction pathway affects the accumulation of functional metabolites in D. officinale

This study showed that blue laser irradiation was the most beneficial among the treatments to promote the accumulation of total flavonoids, polysaccharides, and alkaloids in D. officinale. Based on the study of the blue light signal transduction pathways in model plants such as Arabidopsis (Meng & Lin, 2013; Liu et al., 2013; Wang et al., 2015; Yang et al., 2017), transcription was explored using D. officinale data, the blue laser signal network-related genes were screened, and the blue laser signal network affecting the functional metabolites of D. officinale was initially constructed (Fig. 7A).

Figure 7 (A) Blue laser signal network of functional metabolites in D. officinale. (B) Heat maps of the blue laser signaling networks in up- and downregulated genes.

Red font indicates differentially expressed genes.

Plants can accurately perceive light conditions from UV-B to far-red light through a variety of photoreceptors (Su et al., 2017). The response of plants to blue light is mainly mediated by CRY photoreceptors, including CRY1, CRY2 and CRY-DASH (Yang et al., 2017). In this study, only CRY DASH was found among the DEGs, and the blue laser treatment of Dendrobium_catenatum_newGene_8958 and gene-MA16_Dca024302 (CRY DASH) had the highest expression level, followed by blue light treatment and white light (Fig. 7B). The CRY DASH protein has the biochemical activity of repairing single-stranded DNA and may be involved in the protection of organelle genes. It can directly bind DNA or RNA and can regulate the developmental process by regulating gene transcription, but its signal transduction pathway has not been reported thus far (Castrillo et al., 2015). Therefore, it is speculated that the CRY-DASH receptor plays a leading role in the response of D. officinale to blue laser irradiation.

In the plant COP1/SPA pathway, CRYs can interact with the SPA1/COP1 complex at the posttranscriptional level to indirectly regulate gene expression (Wang et al., 2015). CRY1 inhibits the degradation of HY5 (long hypocotyl 5) and HFR1 (long hypocotyl in Far-Red1) proteins by COP1 under blue light (Yang et al., 2017). CRY2 can also inhibit the degradation of the transcription factor CO protein by COP1 under blue light. These transcription factors regulate the photomorphogenesis of plants (Yang et al., 2017). In this study, only three of these genes, SPA1, HY5, and CO, were found among the DEGs. Blue laser processing gene-MA16_Dca007860 (SPA1), Dendrobium_catenatum_newGene_1002 and gene-MA16_Dca018475 (HY5) had the highest expression, followed by blue laser processing, followed by white light, and gene-MA16_Dca001117 (CO5), gene-MA16_Dca019470 (CO14), gene-MA16_Dca019470 (CO14), and gene-MA16_Dca024960 (CO16) were the opposite (Fig. 7B). Under light conditions, COP1 and SPA can form a complex in the nucleus and regulate the synthesis of plant functional metabolites (Huang, Ouyang & Deng, 2014). HY5 (LONG HYPOCOTYL 5), as a key factor of light signal transduction, also plays an important role in the regulation of plant functional metabolites (Shi et al., 2016). Studies in Arabidopsis have confirmed that blue light can regulate the expression of PAP1 (an R2R3-MYB transcription factor) through the light signal transcription factor HY5 and thereby regulate the metabolic synthesis of flavonoids (Shi et al., 2016). Therefore, it is speculated that SPA1 and HY5 play a positive role in the blue laser-mediated regulation of functional metabolites in D. officinale, while CO plays a negative regulatory role.

In the plant PIF pathway, PIFs (phytochrome-interacting factors) play a key regulatory role in the light signaling pathway (Liu et al., 2019). PIFs can transduce light signals downstream of phytochrome and cryptochrome receptors, integrate hormones, glucose metabolism, circadian rhythm and other signals with environmental pathway signals such as temperature, light quality, light intensity, and photoperiod, and participate in the plant morphogenesis, shade avoidance response, anthocyanin synthesis and carotenoid pigment synthesis and other transcriptional responses (Leivar et al., 2012). In a study of Artemisia annua, it was found that AaPIF3 indirectly regulates artemisinin biosynthesis genes by directly activating the transcription of AaERF1. The content of artemisinin increased with the overexpression of AaPIF3 in transgenic plants. This showed that AaPIF3 plays a significant role in regulating the biosynthesis of artemisinin (Liu et al., 2015). Arabidopsis PIF3 can positively regulate anthocyanin biosynthesis, while PIF4 and PIF5 play a negative regulatory role in red light-induced anthocyanin accumulation. Relevant studies have found that red light effectively increased anthocyanin accumulation in wild-type plants, while the effect in pif4 and pif5 mutant plants was significantly enhanced, and the effect in PIF4OX and PIF5OX overexpression plants was significantly weakened. The transcription levels of the anthocyanin synthesis-related genes CHS, F3′H, DFR, LDOX, PAP1 and TT8 in pif4- and pif5-mutant plants were significantly increased, while those in PIF4OX- and PIF5OX-overexpressing plants were the opposite (Zhang et al., 2019). The blue laser treatment increased expression of Dendrobium_catenatum_newGene_10821 (PIF3) and gene-MA16_Dca021603 (PIF4), while it was lowest under the white light treatment (Fig. 7B). Therefore, it is speculated that PIF3 and PIF4 play an important role in the regulation of functional metabolites of D. officinale by blue laser irradiation.

In the plant MYC2 pathway, MYC2 (a basic helix-loop-helix (bHLH) transcription factor) is a node in the blue light signaling pathway regulating plant metabolism and metabolite synthesis (Sethi et al., 2014). In research on Catharanthus roseus, it was found that MYC2 can promote the accumulation of alkaloids (Gangappa et al., 2013). Arabidopsis MYC2 positively regulates the biosynthesis of flavonoids by positively regulating other transcription factors. In contrast, MYC2 negatively regulates the biosynthesis of the JA-responsive tryptophan derivative indole glucosinolates (Zhang et al., 2011). In this study, only the MYC2 gene was found among the DEGs, and the MA16_Dca025360 and MA16_Dca006997 (MYC2) genes had the highest expression levels under white light, followed by blue light and blue laser light (Fig. 7B). Therefore, it is speculated that MYC2 is a negative regulator of the blue laser irradiation effects on the functional metabolites of D. officinale.

In summary, this study suggests that blue laser can affect the synthesis of functional metabolites of D. officinale through three ways. First, under the action of a blue laser, the interaction between CRYs and the SPA protein inhibits the activation of COP1 by SPA, thereby inhibiting the degradation of HY5 and CO transcription factors by COP1 and upregulating the expression of light-regulated genes, thereby promoting the synthesis of functional metabolites in D. officinale. Second, the blue laser negatively regulates the synthesis of functional metabolites through the transcription factor MYC2. Third, CRYs bind to PIF3 and PIF4, thereby regulating the synthesis of functional metabolites.

Conclusion

This study provides the first demonstration of blue laser irradiation on mRNAs involved in functional metabolites of D. officinale through an RNA-seq analysis. We found that the number of red leaves of D. officinale under blue laser was greater than that under blue and white light. Blue laser had the greatest promoting effect on total flavonoids, anthocyanin, polysaccharides, and alkaloids. Based on the transcriptomic, physiological and biochemical analyses, we revealed that D. officinale responds to blue lasers through cutin, suberine and wax biosynthesis. Heat shock proteins play an important role in the response of D. officinale to blue lasers. The blue laser signal transduction pathway affects the accumulation of functional metabolites in D. officinale. These findings will be helpful for generating new insights for the high-yield production of functional metabolites of D. officinale.

Supplemental Information

Supplemental Information 1 Supplemental information.

Click here for additional data file.

We thank American Journal Experts for editing the English text of a draft of this manuscript.

Additional Information and Declarations

Competing Interests

Author Contributions

DNA Deposition

Data Availability

The authors declare that they have no competing interests.

Hansheng Li conceived and designed the experiments, analyzed the data, prepared figures and/or tables, authored or reviewed drafts of the paper, and approved the final draft.

Yuqiang Qiu performed the experiments, analyzed the data, prepared figures and/or tables, authored or reviewed drafts of the paper, and approved the final draft.

Gang Sun conceived and designed the experiments, prepared figures and/or tables, and approved the final draft.

Wei Ye conceived and designed the experiments, authored or reviewed drafts of the paper, and approved the final draft.

The following information was supplied regarding the deposition of DNA sequences:

All sequencing data of D. officinale under the different light treatments area availble in the National Genomics Data Center (NGDC) Sequence Read Archive: PRJCA006154.

The following information was supplied regarding data availability:

The experiment data and raw data of qRT-PCR are available in the Supplemental File.

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
