# Peer review of "RNA sequencing-based exploration of the effects of blue laser irradiation on mRNAs involved in functional metabolites of D. officinales"

_PeerJ, doi:10.7717/peerj.12684_

## Round 0.1 · original submission · Major Revisions

Dear Dr. Sun and Dr. Ye,

As you can see, our reviewers found that your study was important and interesting, however, they provided several comments and suggestions to strengthen your work.

Reviewer 1 had essential comments for experimental design, the validity of the findings, and the manuscript organization. For example, Table 1 has no lncRNA result. Reviewers 2 and 3 were more critical. For example, there were concerns about the impact and novelty - They were not assessed. In addition, you should add some information on light effect on plant growth and development.

I agree with almost all the comments and suggestions. I would like to
ask you to address or to respond with reasons not to follow the
suggestions made by these reviewers.

Best regards,
Atsushi Fukushima

Reviewer 1 ·

Basic reporting

1. In line 102, the format for units need to be written correctly, such as
“ 1 g L−1 ” “ 6 g L−1 ” need to be revised as “ 1 g·L−1 ” “6 g·L−1” et al.
2. In line 132, “P value” needs to be revised as “P-value”, and in the whole manuscript, please unified the writing, modify “qPCR” as “qRT-PCR”.
3. Table 1 has no lncRNA result in the table and the whole manuscript also has no lncRNA analysis, please explain it or delete it.
4. A uniform format is needed for the “References” part.
5. The legend in Figures 3A-3C was not complete, please explain clearly.

Experimental design

1. Why the author selected the tissue culture seedlings of D. officinale in the incubator for 60 days to sequencing and assay the different functional metabolites.

Validity of the findings

1. In the results, actually, in line 180 “differentially expressed genes” need to be as “differentially expressed mRNA”. And we suggested that this paragraph could merge with line 236. For another, we want to know why to choose the top 10 up- and down-regulated DEG mRNAs analysis under different light treatments. Why didn’t choose specific expression genes to analyze under the blue laser?
2. In line 446, How to establish a blue laser signal network, the discussion part needs to be explained clearly.

Additional comments

1. In line 310, what's the criterion of the 9 groups mRNAs to further verify by qRT-PCR should be mentioned.
2. In the discussion, in line 325, the conclusion is not correct: “D. officinale responds to blue lasers through cutin, suberine, and wax biosynthesis ”, from the manuscript have no direct evidence to prove that, this is a hypothesis, and probably exist, please consider.

Reviewer 2 ·

Basic reporting

Clear and unambiguous

Experimental design

Research question well defined, relevant & meaningful

Validity of the findings

Impact and novelty not assessed.

Additional comments

1.In line 17, when the abbreviated noun “ LB-B, LB-W and W-B” first appears, please mark the full name.
2.In line 157, Please explain the use of the ACTN gene as the reference gene for qRT-PCR.
3.In line 169, “the number of red leaves under the blue laser treatment was greater than that under the blue and white light”, there was a statistical analysis? Please write out the statistics on the number of plants.
4.The focus of the research is not prominent and too protracted, so it is suggested that integrate the expression of differential genes and secondary metabolite contents into one graph for comparative analysis, so that the results will be more intuitive, clear and easy to understand for other readers.

Reviewer 3 ·

Basic reporting

This article presents the D. officinale growth under blue laser, blue and white light. According to the results from RNA-seq, Physiological and biochemical experiment, the author find that blue laser is more efficient in production of metabolites in D. officinale. This research is interesting, however, the paper is poorly written, I suggest the author should make major revision to meet the magazine requirement.

Experimental design

It is well designed

Validity of the findings

The finding is interesting.

Additional comments

1.The introduction is too long, too many information about laser light, I suggest to add some information on light effect on plant growth and metabolites production.
2.Why did the author focus on blue light, how about other light quality including red, green and yellow light on the growth of D. officinale. The paper discuss red light on anthocyanin biosythesis.
3.The author should do correlation between RNA-Seq data and qRT-PCR.
4.The phenotype of D. officinale under difffferent light treatments showed the leave color after blue laser light treatment is much deeper, which may revealed that the anthocyanin is much higher in this treatment. I suggest to assay the anthocyanin content in different treatment.
5.In the discussion part, I suggest the author should put the light effect on plant forward, then discuss on RNA-Seq, and functional metabolic material.
6.The author discuss a lot on ROS and enzymatic and nonenzymatic antioxidant systems including GSH, so I prefer to see those data in the manuscript.
7.I suggest the author should disuss more deeply which combined the RNA-Seq data and the the physiological and biochemical result more closely. Such as POD content and its relationship with the POD biosynthesis genes.
8.The reference of 336-339 is missing or wrong cited, it make me confused.
9.The color of different treatment should be consistent, the white treatment is in green color in the figure 5, but in white light in figure 6.
10.Table 1:have you done IncRNA experiement?
11.Table S2: which does the FR1, FR4 mean?
12.Table S4: is the FPKM vale or log2fold change value?

---

## Round 0.2 · accepted · Accept

Dear Dr. Li and coauthors,

Thank you for your revision. Congratulations.

Best regards

Reviewer 1 ·

Basic reporting

no comment

Experimental design

no comment

Validity of the findings

no comment

Additional comments

no comment

Reviewer 2 ·

Basic reporting

Professional article structure, figures, tables

Experimental design

Methods described with sufficient detail & information to replicate

Validity of the findings

Conclusions are well stated